# Whole Genome Sequencing Reveals Novel Insights about the Biocontrol Potential of *Burkholderia ambifaria* CF3 on *Atractylodes lancea*

**DOI:** 10.3390/microorganisms12061043

**Published:** 2024-05-22

**Authors:** Yongxi Du, Tielin Wang, Chaogeng Lv, Binbin Yan, Xiufu Wan, Sheng Wang, Chuanzhi Kang, Lanping Guo, Luqi Huang

**Affiliations:** 1School of Pharmacy, Nanjing University of Chinese Medicine, Nanjing 210023, China; duyongxi01@gmail.com; 2State Key Laboratory for Quality Ensurance and Sustainable Use of Dao-di Herbs, National Resource Center for Chinese Materia Medica, China Academy of Chinese Medical Sciences, Beijing 100700, China; wtl82@163.com (T.W.); lcgfim@126.com (C.L.); yb51598@126.com (B.Y.); xiufuwan@163.com (X.W.); mmcniu@163.com (S.W.); kangchuanzhi1103@163.com (C.K.); 3Key Laboratory of Biology and Cultivation of Herb Medicine, Ministry of Agriculture and Rural Affairs, Beijing 100700, China

**Keywords:** *Burkholderia ambifaria*, whole genome sequencing, biocontrol, root rot disease, *Fusarium oxysporum*

## Abstract

Root rot caused by *Fusarium* spp. is the most destructive disease on *Atractylodes lancea*, one of the large bulks and most common traditional herbal plants in China. In this study, we isolated a bacterial strain, CF3, from the rhizosphere soil of *A. lancea* and determined its inhibitory effects on *F. oxysporum* in both in vitro and in vivo conditions. To deeply explore the biocontrol potential of CF3, we sequenced the whole genome and investigated the key pathways for the biosynthesis of many antibiotic compounds. The results revealed that CF3 is a member of *Burkholderia ambifaria*, harboring two chromosomes and one plasmid as other strains in this species. Five antibiotic compounds were found that could be synthesized due to the existence of the bio-synthesis pathways in the genome. Furthermore, the synthesis of antibiotic compounds should be confirmed by in vitro experiments and novel compounds should be purified and characterized in further studies.

## 1. Introduction

Cangzhu is the dried rhizome of *Atractylodes lancea* (Thumb.) DC, which is a perennial herbaceous plant belonging to Compositae. It was first recorded in an ancient Chinese medical classic, “Shennong’s Herbal Classic”, and it has the effects of drying dampness, invigorating the spleen, dispelling wind, dispelling cold, and brightening the eyes [1]. Cangzhu is a major medicinal material in China and also plays an important role in the medicinal material markets in East Asia countries [2,3]. In recent years, with the continuous expansion of artificial planting areas and the increases in planting years, the problem of *A. lancea* disease caused by *Fusarium oxysporum* has become increasingly prominent, and it has become one of the key factors hindering the development of the Cangzhu industry. Especially in various regions, the soil-borne fungal disease of *A. lancea,* root rot, is particularly serious [4]. It not only has a significant impact on the yield of *A. lancea* but also poses a threat to the quality of plants intended for medicinal use. *F. oxysporum* is an extremely common saprophytic fungus widely distributed in various soil types and climatic conditions. Although most species are harmless to plant growth, some species are highly pathogenic and can cause severe vascular wilt diseases in food and economic crops. These diseases not only affect crop yield but also lead to decreases in quality, thus causing billions in economic losses for global agriculture each year [5,6,7]. When lacking host plants, *F. oxysporum* can grow and reproduce in plant residues and organic matter, with high adaptability and survival ability. It can also survive in the soil for up to 10–15 years by forming dormant spores, and it remains active even under harsh environmental conditions [7,8,9,10]. This long-term dormancy allows *F. oxysporum* to rapidly recover and infect new host plants under appropriate conditions. What is more troublesome is that once the pathogen *F. oxysporum* colonizes in soil through infected plants, it is extremely difficult to completely eliminate [11,12]. Due to the presence of dormant spores and its strong survival ability, it is still difficult to completely eradicate even with various control measures such as chemical fungicides and biological control [13]. Once root rot breaks out, the growth of *A. lancea* is blocked and the effective components of the medicinal materials are reduced, and thus, the efficacy of Cangzhu as a medicinal material is affected. Therefore, the prevention and treatment of root rot disease has become an urgent task for the current development of the *A. lancea* industry.

As a green and environmentally friendly prevention and treatment method, biological control has gradually been applied in the *A. lancea* industry. Antagonistic microorganisms are a major weapon for controlling root rot [14]. These microorganisms can inhibit the growth and reproduction of pathogenic bacteria by competing for nutrients with pathogenic bacteria, occupying ecological niches, or producing antibacterial substances, thereby reducing the impacts of diseases on *A. lancea*.

Soil-borne beneficial microorganisms represent a significant component of agricultural biocontrol agents. To date, numerous biocontrol agents isolated from soil have been effectively employed to manage root rot disease in agricultural settings, including *Trichoderma* spp., *Streptomyces* spp., and *Bacillus* spp. [15,16]. The genus *Burkholderia* has become increasingly important over the past several decades because of its ability to produce abundant secondary metabolites with antimicrobial, insecticidal, herbicidal, or growth-promotion traits [17,18,19,20]. Several *Burkholderia* strains, such as *B. vietnamiensis*, *B. ambifaria*, and *B. pyromania*, have been reported to be potential and efficient biological control agents [21,22,23]. The antibacterial secondary metabolites cepacin, phenazine, and pyrrolnitrin produced by *Burkholderia* are thought to be critical mechanisms of resistance to plant pathogens [24]. *Burkholderia* sp. HQB-1 isolated from the inter-rhizosphere soil of banana plants can protect bananas from Fusarium wilt through the secretion of phenolic-1-carboxylic acid [25,26,27,28]. *Burkholderia* has not only biocontrol activity but also plant growth-promoting activity. *B. ambifaria* T16 has been shown to inhibit several *Fusarium* species’ growth and promote biomass increases in barley plants [29,30]. Therefore, *Burkholderia* could be an essential resource for bio-fungicides or biofertilizers for agriculture.

This study aimed to isolate and investigate a bacterium from *A. lancea* rhizosphere soil with potent inhibitory effects on *F. oxysporum*. Upon sequencing and identification, the bacterium was identified as *B. ambifaria*, which we designated as strain CF3. Our subsequent research delved into the antibacterial and biocontrol properties of CF3, revealing its inhibitory impact on pathogenic fungi. Notably, we demonstrated the efficacy of CF3 in controlling *A. lancea* root rot, demonstrating its preventive and curative capabilities under greenhouse conditions. The discovery of *B. ambifaria* CF3 as a novel strain for managing *A. lancea* root rot offers a promising avenue for sustainable agriculture and biological control, providing a valuable resource for microbial-based plant disease management.

## 2. Materials and Methods

### 2.1. Sample Collection and Bacterial Isolation

Three rhizosphere soil samples were collected from a five-year *A. lancea* field that was infected by root rot pathogens in Liaoyang City, Liaoning Province (41°15′52.0″ N, 123°08′42.9″ E). The samples were taken from healthy plants using the methods described in [25,29]. We took 13 g of soil from the roots and surrounding area of each plant and mixed it with 100 mL of sterile PBS buffer (containing 137 mM NaCl, 2.7 mM KCl, and 10 mM KH_2_PO_4_, with a pH of 7.4). First, the samples were shaken in an ultrasonic bath for 10 min, followed by shaking at 150 rpm for 30 min to create a rhizosphere suspension. Subsequently, the suspension was 10-fold diluted 5 times and inoculated onto LB solid medium containing 50 μg/mL of nystatin. The plates were incubated at 30 °C for 2 days. Colonies with different morphological characteristics were selected and inoculated onto fresh LB agar, and this process was repeated until a uniformly distributed population of colonies was obtained. For long-term storage, the strains were inoculated into liquid medium containing 50% glycerol and stored at −80 °C.

### 2.2. Determination of Antifungal Activity In Vitro

We selected eight plant pathogenic fungi, including *F. oxysporum*, *F. solani*, *Colletotrichum gloeosporioides*, *Epicoccum sorghinum*, *Alternaria alternata*, *Rhizoctonia solani*, *Botrytis cinerea*, and *Botrytis elliptica*. These fungal species were isolated and preserved in our laboratory (China Medicinal Plant Pathology Laboratory, Chinese Academy of Sciences). Traditional inoculation methods were used to evaluate antifungal activity. A 5-mm-diameter circular inoculation area of the pathogenic fungus was placed at the center of the PDA plate. On both sides of the disk, 10 μL of bacterial suspension were inoculated along a straight line 2.5 cm away from the disk, ensuring that the bacterial trace was parallel to the fungal disk. Those aside the fungal disk where nothing was placed were used as controls. After incubation at 28 °C for 5–7 days, the distance from the edge of the fungal colony to the bacterial disk, which was the diameter of the inhibition zone, was measured. The percentage of radial growth inhibition (PIRG) was calculated using the following formula: PIRG = [(C − T)/C] × 100%, where C is the radius of the fungal mycelium growth in the control group and T is the radius in the treatment group [31]. All experimental strains were performed in triplicate.

### 2.3. Biocontrol Effect of B. ambifaria CF3 on the Root Rot of A. lancea under Greenhouse Conditions

*A. lancea* seeds were surface-sterilized with 5% sodium hypochlorite for 20 min and rinsed 3 times with sterile distilled water. Pots (7 × 7 × 10 cm) were filled with sterile soil substrate (nutrient soil: vermiculite = 1:1, 100 g), and 25 mL of CF3 cell suspension (10^8^ cfu/mL) was added to the soil of each pot. Seven days after CF3 treatment, seedlings of *A. lancea* with uniform leaf growth were transplanted into the pots. Pathogen inoculation was performed by watering each plant with 25 mL of conidial suspension (10^8^ conidia mL^−1^). Uninoculated control seedlings treated only with sterile distilled water were used as negative controls, while positive controls were inoculated with *Fusarium oxysporum* (Fo) and treated with sterile distilled water [32]. All *A. lancea* plants were grown in a growth chamber with a temperature of 26 °C and a 16 h photoperiod, and they were watered with sterile distilled water every 3 days. Five weeks after inoculation, all parameters were evaluated. Each treatment included 20 seedlings of *A. lancea*, and the experiment was repeated 3 times.

### 2.4. Genome Sequencing, Assembly, Annotation, and Bioinformatics Analysis

In the initial phase of the genomic analysis, the genetic material of CF3 was extracted through the Promega’s Wizard^®^ DNA Purification System, adhering strictly to the manufacturer’s standardized procedures [33]. Subsequently, we employed a TBS-380 Fluorometer (Turner BioSystems, Sunnyvale, CA, USA) to precisely quantify the concentration of the purified DNA, ensuring its quality. To facilitate subsequent experiments, the selected DNA necessitated high purity, with an optimal OD260/280 ratio of 1.8 to 2.0 and a concentration of 286 μg/mL. Through the TBS-380 validation, we confirmed the suitability of the DNA for continuous and optimized scientific research endeavors.

#### Sequencing, Assembly, and Annotation

The whole genome of CF3 was sequenced using a combination strategy that employed an Illumina PE platform and PacBio RS II single-molecule real-time sequencing (Majorbio Bio-Pharm Technology Co., Ltd., Shanghai, China).

### 2.5. Phylogenetic Analysis

The 16S ribosomal RNA gene sequence was subjected to a thorough search in the NCBI GenBank nucleotide database through BLAST analysis for comprehensive sequence retrieval. Alignment of the obtained sequences was performed using the Muscle algorithm incorporated in TBtools-II version 2.08 for multiple sequence alignment purposes [34]. Following this, a phylogenetic tree was constructed employing the MLtree plugin within TBtools-II, version 2.08 [34].

## 3. Results

### 3.1. Antagonistic Activity of B. ambifaria CF3 against Fo In Vitro

We obtained 27 pure bacterial cultures from 10 rhizosphere samples of *A. lancea*. To assess the inhibition activity of these bacteria against plant pathogens, an in vitro antagonistic experiment was performed, and the inhibition rates were also calculated. After 7 days of incubation, only the strain CF3 showed antagonistic activity, while the other stains were not able to inhibit the pathogenic fungi. We observed that the pathogenic fungi grew with strain CF3 and showed the same color and hyphae, but we observed oval colony shapes with the blank control, indicating that the growth of the tested pathogenic fungi was inhibited by CF3. To quantitively evaluate the inhibitory activity of CF3, the inhibition rates were also calculated. Among the tested fungi, *B. cinerea* was inhibited the most, with an inhibition rate of 64.94% while *F. solani* had the lowest inhibition rate of 36.36% (Figure 1). This result indicated that CF3 had the best inhibitory activity against *B. cinerea*, which causes severe root rot disease on *A. lancea*.

### 3.2. Antagonistic Activity of CF3 against F. oxysporum in Planta

After 2 weeks of incubation, the *A. lancea* seedlings inoculated with Fo showed typical symptoms and the whole plants were wilted, while the *A. lancea* seedlings inoculated with sterile water (Water) remained green and healthy (Figure 2). The *A. lancea* seedlings inoculated with *B. ambifaria* CF3 and *F. oxysporum* (Fo + Ba) showed green leaves with few wilt symptoms and fewer roots than those that received the Water treatment (Figure 2). These results suggested that the CF3 strain had antagonistic activity against *F. oxysporum* under greenhouse conditions.

To evaluate the protective effect of CF3 on *A. lancea* seedlings, the biological characteristics of the whole plants were measured. The average shoot height of the Fo-treated plants was 6.74 cm, significantly lower than that of the Water- and Fo + Ba-treated plants, had average shoot heights of 15.71 cm and 13.47 cm, respectively (Figure 3C). The average root length of the Fo-treated plants was 1.58 cm, and this was also significantly lower than that of the Water- and Fo + Ba-treated plants, which had average root lengths of 9.48 cm and 6.47 cm, respectively (Figure 3F). The average fresh and dry weights of the Fo-treated plants were 1.05 g and 0.139 g, respectively, which were significantly lower than those of the Water- and Fo + Ba-treated plants, which were similar to each other (Figure 3A,B). To evaluate the effect of CF3 on the roots of *A. lancea*, the root fresh and dry weights were also measured. The average fresh and dry weights of the Fo-treated plants were 0.396 g and 0.097 g, respectively, which were significantly lower than those of the Water- and Fo + Ba-treated plants (Figure 3D,E). This result indicated that *F. oxysporum* could destroy the *A. lancea* seedlings, while *B. ambifaria* CF3 could relieve the root rot symptoms, indicating the in vivo inhibitory activity against the pathogen *F. oxysporum.*

### 3.3. Genome Sequencing of B. ambifaria CF3

To gain a deep insight into the phylogeny and function of the *B. ambifaria* CF3 strain, its whole genome was sequenced and assembled. Our study employed a dual strategy, utilizing the PacBio RSII platform to construct a third-generation sequencing library. This method facilitated the efficient and precise whole-genome sequencing of the strains, enabling in-depth investigation of their disease-resistance mechanisms. By integrating the Illumina PE and PacBio RS II single-molecule real-time technologies, we obtained a total of 302,071 high-fidelity reads that were characterized by an average length of 5738 base pairs and an N50 value of 10,898 base pairs (Table 1). After the genome was assembled, the complete genome size was 7,573,820 bp, consisting of 2 circular chromosomes and a plasmid. The first chromosome was 4,682,183 bp, with a GC content of 66.73%; the second chromosome was 2,651,331 bp, with a GC content of 66.89%; and the plasmid was 240,306, with a GC content of 61.28% (Figure 4). Through coding gene-prediction, a total of 6829 protein-coding genes were predicted, accounting for 85.52% of the genome, including 18 rRNA genes, 67 tRNA genes, 1 sRNA, 15 gene islands, 13 phages, and 15 CRISPRs. The protein-coding genes in the NR, SwissProt, Cluster of Orthologous Groups of proteins (COG), KEGG, Pfam, and Gene Ontology (GO) databases are 1786, 2761, 5352, 6511, 4514, and 4514, respectively.

#### 3.3.1. Phylogenetic Analysis Revealed That the CF3 Strain Belongs to *Burkhoderia ambifaria*

To deepen our understanding of the phylogenetic context of *B. ambifaria* CF3, a comprehensive phylogenetic tree was constructed based on 16S ribosomal RNA (rRNA) sequences (Figure 5). The resulting analysis confirmed that the CF3 strain was placed within the broader *Burkholderia* genus, clustering closely with *B. ambifaria* AMMD, thereby reinforcing its classification within this bacterial family.

#### 3.3.2. ANI Analysis Revealed That the CF3 Strain Is a Member of *Burkholderia ambifaria*

Average nucleotide identity (ANI) is a powerful approach for evolutionary distance assessments between bacteria, and therefore, we employed this method to identify the CF3 at the species level. The ANI values of *B. ambifaria* CF3 between *B. ambifaria* CP066037.1, *B. ambifaria* CP092843.1, and *B. ambifaria* CP009798.1 were 97.70%, 97.59%, and 97.70%, respectively, all of which were greater than 96%, which was considered as a criterion that distinguished a species (Figure 6). According to this result, we identified the CF3 strain as a strain of *B. ambifaria*.

#### 3.3.3. Analysis of CAZyme Genes for the *B. ambifaria* CF3 Genome

A comprehensive analysis identified 224 CAZymes, consisting of 81 glycoside hydrolases (GH), 112 glycosyltransferases (GT, primarily GT2 with 112 members), 11 carbohydrate esterases (CE), 35 carbohydrate-binding modules (CBM), 2 polysaccharide lyases (PL), and 9 auxiliary activities (AA). Among these enzymes, β-glucanases (GH1), β-glucosidases (GH3), chitinases (GH18 and GH19), cellulases (GH5), endoglucanases (GH5 and GH51), trehalases (GH68), and specific starch-debranching enzymes (GH13_31, GH13_28, and GH13_29) as well as L-fructosyltransferase (GH32) from the GH family displayed potential antifungal properties (Figure 7). This finding underscored the crucial role of CAZymes in CF3’s antifungal defense mechanism.

The synthesis pathways of secondary metabolites that had antibiotic activities were also predicted by our genomic analysis. The terpene pathway had the most clusters and genes while other compounds had one or two synthesis clusters (Figure 8).

## 4. Discussion

Root rot caused by *Fusarium* spp. is the largest barrier to the production of *A. lancea*, which maintains economies in many rural places in China. We isolated a bacterial strain, CF3, that showed inhibitory activity to *F. oxysporum* and *F. solani*, and we confirmed its control effect in greenhouse conditions (Figure 1, Figure 2 and Figure 3). Additionally, we sequenced the genome of the strain CF3 and identified it as a strain of *B. ambifaria*, and we also explored the synthesis pathways involved in its antifungal activity (Figure 4, Figure 5 and Figure 6).

*F. oxysporum* is considered as the main causal agent of root rot or fusarium wilt on many plants, including many traditional Chinese medicinal plants. In this study, the *A. lancea* seedlings inoculated with *F. oxysporum* were almost completely destroyed, while those inoculated with *F. oxysporum* and CF3 grew out and showed green leaves, indicating the preventive effect of CF3 as a biocontrol agent (Figure 2). 

*Burkholderia ambifaria* was proposed as a new member of the *Burkholderia cepacia* complex in 2001, and its model strain is *B. ambifaria* AMMD LMG 19182 [35]. Since then, the *B. ambifaria* strains have been studied and developed as biological control agents, as well as bio-enzyme, vaccine, and natural production agents. The genomes of the *B. ambifaria* strains were comprised of three chromosomes/replicons in which the smallest chromosome (the third replicon) possessed the greatest variation in sequence capacity, whereas the other chromosomes were more consistent in size. In this study, we obtained two chromosomes with sizes of 4.68 Mb and 2.65 Mb, respectively. Meanwhile, we also identified a chromosome, or so-called third replicon, with a size of 0.24 Mb, which we called the plasmid in this study. 

*Burkholderia* spp. have a great potential for bio-technological applications in plant growth promotion as the biological control of important soil-borne phytopathogenic fungi requires rich antibiotic synthesis genes [36], and it has been reported to produce a large number of antifungal substances such as pyrrolnitrin, siderophores, and phenazines [37], all of which play important roles in controlling fungal diseases in plants. In this study, we found the synthesis pathways of antibiotic compounds, such as terpene, arylpolyene, lanthipeptide, T1PKS, and butyrolactone, in the genome of *B. ambifaria* CF3, suggesting its possible broad anti-fungal spectrum (Figure 8).

Approximately 300 antibacterial and/or antifungal terpenes have been identified so far [38]. Phosphonates are systemic fungicides widely employed in agriculture, such as Fosetyl-Al and various phosphonate salts such as potassium, sodium, or ammonium phosphonates [39]. Lanthipeptides are peptides synthesized post-translationally by ribosomes, and they are characterized by the presence of lanthionine bridges. Lanthipeptides with antibacterial activities are known as lantibiotics. Their antibacterial mechanism is unique, and the targeted bacteria find it difficult to develop resistance, making them a crucial element in the development of new antibiotic drugs [40]. Butyrolactone, as a potential fungicide, has significant fungicidal activity against *C. anthracis* and *B. cinerea* [41].

In this study, we investigated the profound influence of Fusarium root rot on *A. lancea,* centering on identifying the beneficial role of the strain CF3 in disease management. The strain CF3 was classified as *B. ambifaria,* and a genomic analysis revealed a diverse array of antifungal pathways, such as terpenoids, polyenes, wool-like thiol peptides, T1PKS, and butenolides. These findings suggest that *B. ambifaria* CF3 exhibits a broad spectrum of antimicrobial properties, potentially positioning it as a promising biocontrol agent, particularly against *F. oxysporum* and *F. solani*.

The study validated CF3’s efficacy in greenhouse settings and contributed to theoretical groundwork for novel biopesticides and biocontrol techniques. The *Burkholderia* genus, which includes *B. ambifaria*, has significant biotechnological potential due to its natural ability to produce antibiotics and combat fungi. This text highlights the relevance of CF3 in sustainable agriculture and plant-health management. Future research can improve its application in agriculture by refining cultivation conditions and exploring its potential for controlling additional plant diseases. Developing a deeper understanding of the antifungal mechanisms of CF3 will help create more effective and environmentally friendly biocontrol measures. This will reduce dependence on chemical pesticides and promote environmental and farmer health.

## Figures and Tables

**Figure 1 microorganisms-12-01043-f001:**
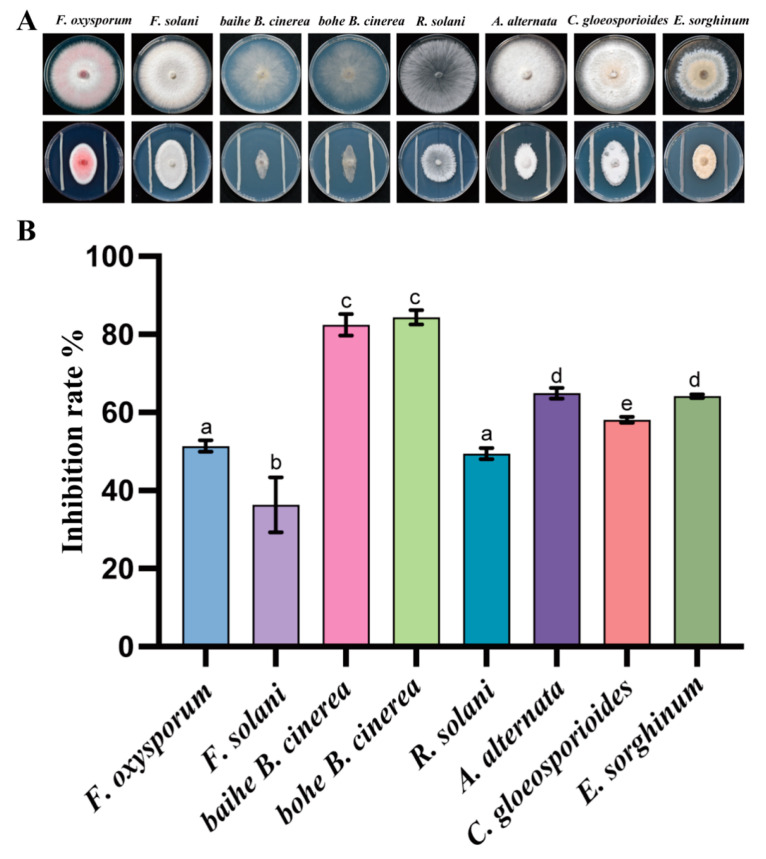
Antagonistic activity of CF3 against Fo in vitro. The growth inhibitory effect of CF3 on different pathogenic bacteria in vitro (**A**) and its antibacterial rate (**B**). a–e indicate differences among samples. Different letters represent significant differences (*p* < 0.05), while the same letters indicate no significant differences.

**Figure 2 microorganisms-12-01043-f002:**
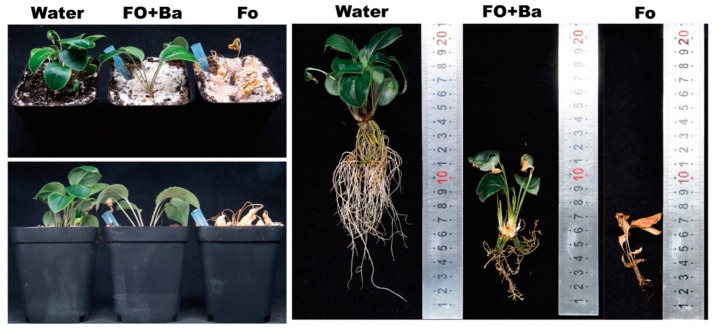
Antagonistic activity of CF3 against *F. oxysporum* in Planta.

**Figure 3 microorganisms-12-01043-f003:**
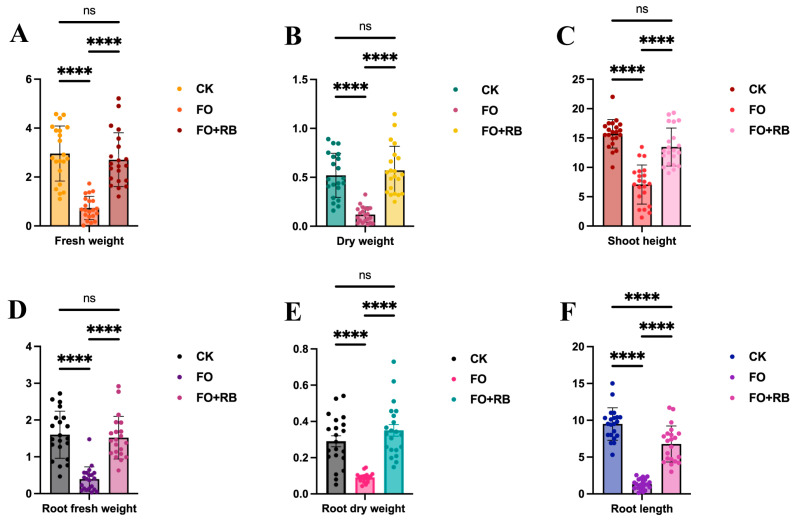
The quantitative analysis of the antagonistic activity of CF3 against *F. oxysporum*. (**A**,**B**) The fresh and dry weights of the whole *A. lancea* seedlings. (**D**,**E**) The fresh and dry weights of the roots of the *A. lancea* seedlings. (**C**,**F**) The shoot and root lengths of the *A. lancea* seedlings. ‘ns’ denotes no significant difference (*p* > 0.05), and ‘****’ denotes extremely significant differences (*p* < 0.0001).

**Figure 4 microorganisms-12-01043-f004:**
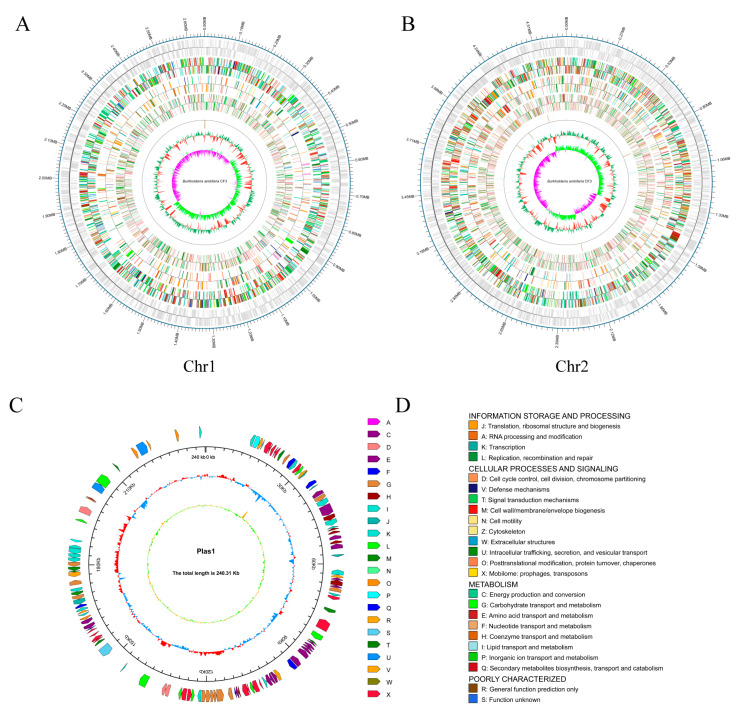
The genomic map of *B. ambifaria* CF3. (**A**) Chromosome 1. (**B**) Chromosome 2, displaying the coding genes from the outer to inner regions, followed by the gene function annotations (COG, KEGG, and GO) and the non-coding RNAs, respectively. (**C**) The plasmid map. From the outer to the inner layers, the clockwise arrows represent the COG functional annotation genes (coding strand), genomic sequence coordinates, GC content, and GC skew. (**D**) Legend for the COG database annotations.

**Figure 5 microorganisms-12-01043-f005:**
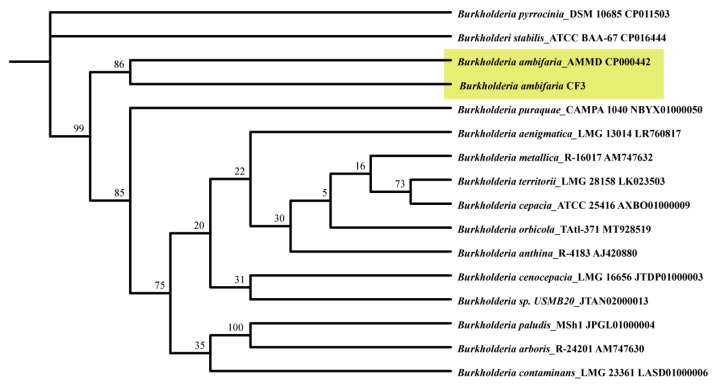
A phylogenetic dendrogram was derived from a comparative analysis of 16S ribosomal RNA (rRNA) gene sequences, revealing the evolutionary lineage of *B. ambifaria* CF3 within its closely related *Burkholderia* taxa. The reliability of the branching patterns was assessed through Bootstrap values, represented as percentages after 1000 replicates, and each node was annotated with the respective GenBank accession numbers. The ML phylogenetic tree was computationally constructed by employing the TBtools-II software (v2.096) package.

**Figure 6 microorganisms-12-01043-f006:**
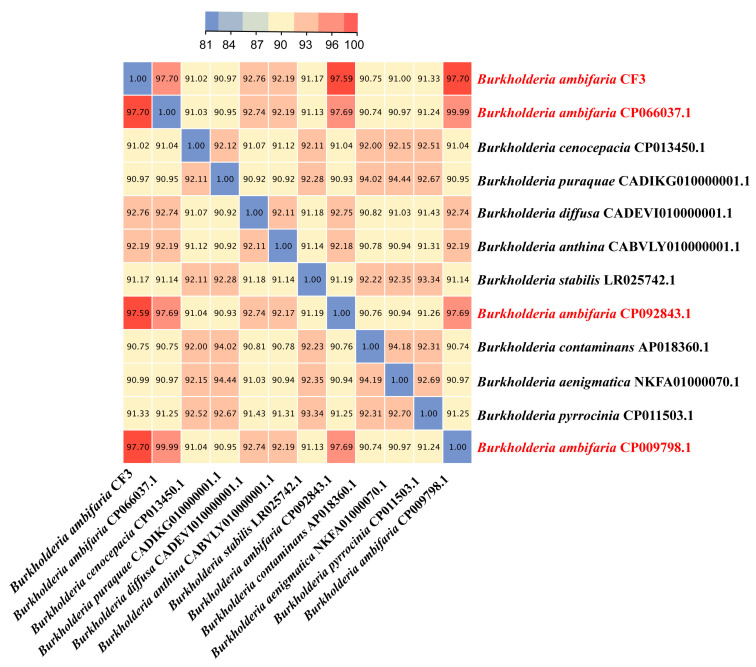
A heatmap was generated to illustrate the pairwise average nucleotide identity (ANI) values for the complete genomes of *B. ambifaria* CF3 and a comparative set of five other *Burkholderia* species, providing a visual representation of the genetic similarity among these bacterial strains.

**Figure 7 microorganisms-12-01043-f007:**
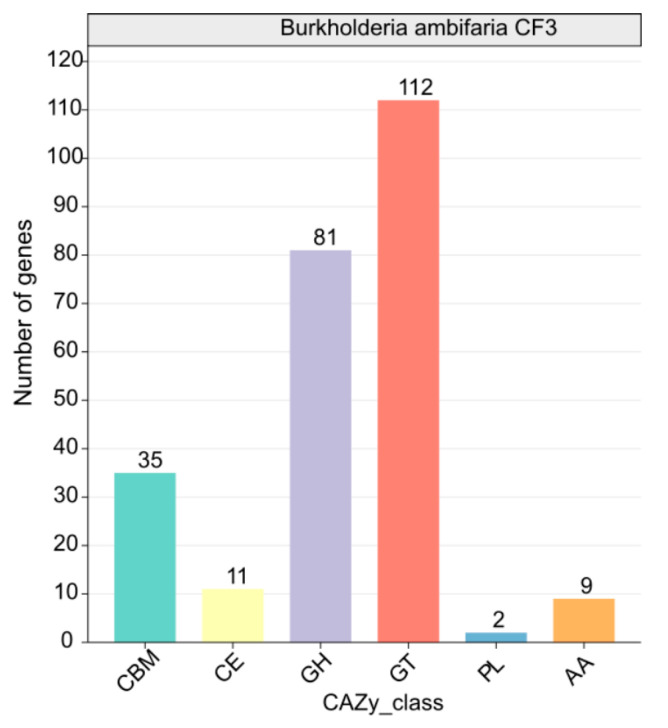
Functional characterization of the glycoside hydrolase family is based on the CAZy database.

**Figure 8 microorganisms-12-01043-f008:**
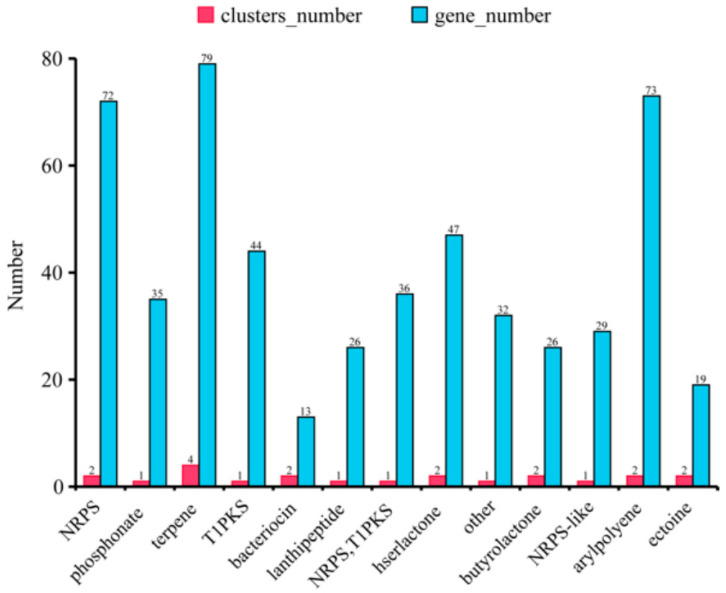
The hypothetical gene cluster lists encoding the secondary metabolites predicted by antiSMASH in the genomes of *B. ambifaria* CF3.

**Table 1 microorganisms-12-01043-t001:** Genome characteristics of the *B. ambifaria* CF3 strain.

Characteristics	Value
Genome size (bp)	7,573,820
GC content (%)	66.89
Topology	Circular
Chromosome	2
Chromosome size (bp)	7,333,514
Plasmid	1
tRNA	67
rRNA (5S, 16S, and 23S)	18
sRNA	1
Protein-coding genes (CDS)	6829
Repetitive sequence	96
CRISPR	15
Genomic islands	15
Prophage	15
Gene cluster	13
Genes assigned to NR	1786
Genes assigned to GO	4514
Genes assigned to KEGG	6511
Genes assigned to COG	5352
Genes assigned to Pfam	4514
Genes assigned to Swiss-Prot	2761

## Data Availability

The raw data supporting the conclusions of this article will be made available by the authors on request.

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
