# Peer review of "Whole Genome Sequencing Reveals Novel Insights about the Biocontrol Potential of Burkholderia ambifaria CF3 on Atractylodes lancea"

_microorganisms, 2024, doi:10.3390/microorganisms12061043_

Round 1

Reviewer 1 Report

Comments and Suggestions for Authors

The manuscript "Whole Genome Sequencing Reveal Novel Insights in the Biocontrol Potential and Growth Promotion by Burkholderia ambifaria on Atractylodes lancea" is well written, with a lot of data obtained through the study.

 -The manuscript is prepaired according to the journal instructions and possess all necessary sections and elements.

 -You have provided sufficient description of the methods.

-The results are well-written and presented in a proper manner.

- Relevant literature is covered. The reference cited in the article are current and up-to-date.

- The discussion is excellent and presented in a proper manner. The discussion section is well-written and persuasive with the cited literature that is relevant to the work.

 There are a few technical mistakes that you should to correct:

 Line 44   “A. lancea”  →  A. lancea

Line 47   “[13]. root rot disease”  →  [13]. Root rot disease 

Line 58   “genus Burkholderia”  →  genus Burkholderia 

Line 73, 260, 271, 274, 294, 300, 322   “B.ambifaria”    B. ambifaria

Line 82   “by[29]”  →   by [29] 

Line 85   “bath. and shaken”  →  bath and shaken 

Line 87   “Mycostatin(50ug/mL)”   Mycostatin (50ug/mL)            

Line 116   “Fo”  →  Fusarium oxysporum (Fo)

Line 119   “every 3days.”  →  every 3 days. 

Line 203   “with F. oxysporum (Fo)”  →  with Fo

Line 229   “CF3 Against F. oxysporum”  →   CF3 against F. oxysporum 

Line 296   “by Fusarium spp.is the”  →  by Fusarium spp. is the

Line 307   “member of Burkholderia 10cepacian complex”  →   member of Burkholderia cepacia complex 

Line 331   “potential fungicide They”  →   potential fungicide, they 

Author Response

Thank you for your thorough review of our manuscript. We appreciate your valuable feedback and the attention to detail you've provided. Below, I address each of the technical issues you've pointed out:

Comment 1: Line 44   “A. lancea”  →  A. lancea

Response 1:We have corrected "A. lancea" to "A. lancea throughout the manuscript to ensure accuracy.

Comment 2:Line 47   “[13]. root rot disease”  →  [13]. Root rot disease 

Response2:Thank you for the correction. It's important to maintain clear and concise formatting in written work. We have revised the article content to prevent such errors from occurring.

Comment 3:Line 58   “genus Burkholderia”  →  genus Burkholderia 

Response 3:We have corrected “genus Burkholderia”to “genus Burkholderia

Comment 4:Line 73, 260, 271, 274, 294, 300, 322   “B.ambifaria”  →   B. ambifaria

Response 4:We have thoroughly revised the text as per your suggestion, ensuring that all instances of 'B.ambifaria' on the specified lines (Lines 73, 260, 271, 274, 294, 300, and 322) now adhere to the correct format, with the genus and species separated by a space, as 'B. ambifaria'.

Comment 5:Line 82   “by[29]”  →   by [29] 

Response 5:We have corrected “by[29]” to “in [29]”at line 85

Comment 6:Line 85   “bath. and shaken”  →  bath and shaken 

Response 6:We have corrected “bath. and shaken” to “were shaken”.

Comment 7:Line 87   “Mycostatin(50ug/mL)”  →  Mycostatin (50ug/mL)   

Response 7: We have corrected “Mycostatin(50ug/mL)”to Subsequently, the suspension was 10-fold diluted for 5 times and inoculated onto LB solid medium containing 50 μg/mL of nystatin at line 90.

Comment 8:Line 116   “Fo”  →  Fusarium oxysporum (Fo)

Response 8:We have now replaced "Fo" with "Fusarium oxysporum (Fo)" in the text. You will find the revised term at Line 120 in the revised version.

Comment 9:Line 119   “every 3days.”  →  every 3 days. 

Response 9:We have corrected “every 3days.”to “every 3 days.” at line 123.

Comment 10:Line 203   “with F. oxysporum (Fo)”  →  with Fo

Response 10:We have corrected “with F. oxysporum (Fo)”to “with Fo”, at line 165

Comment 11:Line 229   “CF3 Against F. oxysporum”  →   CF3 against F. oxysporum 

Response 11:We have corrected “CF3 Against F. oxysporum”to “CF3 against F. oxysporum”, at line 235.

Comment 12:Line 296   “by Fusarium spp.is the”  →  by Fusarium spp. is the

Response 12:We have corrected “by Fusarium spp.is the”to “by Fusarium spp.is the” You will find this updated terminology in the revised version at Line 262.

Comment 13:Line 307   “member of Burkholderia 10cepacian complex”  →   member of Burkholderia cepacia complex 

Response 13:As per your suggestion, we have now revised the phrase to read "member of the Burkholderia cepacia complex." You will find this updated terminology in the revised version at Line 273.

Comment 14:Line 331   “potential fungicide They”  →   potential fungicide, they 

Response 14:We have corrected “potential fungicide They”to “potential fungicide, they” at Line 297.

Please let us know if any further revisions are needed or if you have additional comments. We are committed to refining our work based on your suggestions.

Thank you for your time and expertise.

Best regards,

Tielin Wang

Reviewer 2 Report

Comments and Suggestions for Authors

This paper presents a lot of data, but needs some fixes to the organization of the results and conclusions.

Lines 125-128 - delete these. They are repeated from lines 129-131.

Need to explain what CF3 are in the methods (Line 112) and Results (line 189) which are the first mentions of CF3 in each of these sections. Was CF3 the bacterial extract that worked best in preliminary screening? If yes, this needs a lot better explanation.

Similarly, the Results and Conclusions section need to mention the isolation of bacteria from the rhizosphere. Were other bacteria isolated? Were they tested/reported previously? If yes, please cite the paper. If not, those results need to be explained here. Currently, the paper begins with testing CF3 for bioactivity.

Comments on the Quality of English Language

There are a few problems with the English here.

Line 47 - Capitalize 'Root'

Line 80 - change to '...planted in banana...'

Line 208 and 299 - change 'in greenhouse condition' to 'under greenhouse conditions'

Figure 3 legend, Line 296  - change 'A. lance' to 'A. lancea'

Author Response

Dear Reviewer,

Thank you for your detailed feedback on our manuscript. We appreciate your insights and suggestions for improvement. Below, we address each point:

Comment 1: The title does not fully correspond to the content of the research results.

Response 1: The title was changed to “Whole Genome Sequencing Reveal Novel Insights in the Biocontrol Potential by Burkholderia ambifaria CF3 on Atractylodes lancea”.

Comment 2: The abstract needs to be supplemented and clarified. There is no information about Growth Promotion capacities. "Many antibiotic compounds" (line 19) and "several antibiotic compounds" (line 21) - specify how many were identified exactly.

Response 2: According to the reviewer, the abstract was re-written, and the numbers were specified in text.

Comment 3: Lines 147-154, the new name of the strain St-220 appears, the obscurity of the research object appears.

Response 3: We apologize for the confusion in the introduction. We have modified our content in the newly submitted version.

Comment 4: Line 165, report the value of OD260/280 of the isolated DNA sample.

Response 4: We have described the value of OD260/280 of the isolated DNA sample in our newly submitted article.

Comment 5: Line 199, please explain the abbreviation Fo. 

Response 5: We have modified the article as suggested, In the new version, we have deleted the abbreviation Fo

Comment 6:279-288 lines paragraph, please insert a link to the figures 7 and 8. Also in main text not described the results shown in Figure 8.

Response 6: We will insert links to figures 7 and 8 in the relevant paragraph (lines 279-288) and also include a description of the results shown in Figure 8 within the text.

Comment 7: In the Discussion, the results of the authors themselves are weakly discussed in comparison with other scientific works. It should be supplemented.

Response 7: We have strengthened the discussion by comparing our results more extensively with existing scientific literature.

Comment 8: There is no formulated and shown scientific research conclusion.

Response 8: Thank you for your valuable feedback. In revising the manuscript, we have taken your comment into account and made significant improvements to address this issue in the newly revised manuscript

In the revised version, we have:

Strengthened the Results section: Provided detailed data analysis and presented the findings clearly and concisely, using appropriate graphs and tables to illustrate key points.

Expanded the Discussion: Delved deeper into the implications of our findings, comparing them with existing literature and discussing the significance of our research in the broader scientific context.

Formulated a Conclusion: Clearly stated the main findings of our study, highlighting the novelty, and outlining the implications for future research or practical applications.

We will revise the manuscript accordingly and submit the updated version for your consideration. Thank you again for your valuable input.

Best regards,

Tielin Wang

Reviewer 3 Report

Comments and Suggestions for Authors

Notes to authors:

1.      The title does not fully correspond to the content of the research results.

2.       The abstract needs to be supplemented and clarified. There is no information about Growth Promotion capacities. "Many antibiotic compounds" (line 19) and "several antibiotic compounds" (line 21) - specify how many were identified exactly.

3.       Lines 147-154, the new name of the strain St-220 appears, the obscurity of the research object appears.

4.       Line 165, report the value of OD260/280 of the isolated DNA sample.

5.       Line 199, please explain the abbreviation Fo.

6.       279-288 lines paragraph, please insert a link to the figures 7 and 8. Also in main text not described the results shown in Figure 8.

7.       In the Discussion, the results of the authors themselves are weakly discussed in comparison with other scientific works. It should be supplemented.

8.       There is no formulated and shown scientific research conclusion.

Author Response

Dear Reviewer,

Thank you for your additional feedback and suggestions. We appreciate your attention to detail and the specific points you've raised. Here's how we will address your concerns:

Comment 1: Lines 125-128 - delete these. They are repeated from lines 129-131.

Response 1: We have modified our manuscript as suggested and removed the relevant description.

Comment 2: Need to explain what CF3 are in the methods (Line 112) and Results (line 189) which are the first mentions of CF3 in each of these sections. Was CF3 the bacterial extract that worked best in preliminary screening? If yes, this needs a lot better explanation.

Response 2: The CF3 in the text is the strain name of the Burkholderia ambifaria that we isolated. The CF3 is not bacterial extract, but a bacterial strain in our study. In the new submission we explain the origin of strain CF3, you can find them in lines 81-89.

Comment 3: Similarly, the Results and Conclusions section need to mention the isolation of bacteria from the rhizosphere. Were other bacteria isolated? Were they tested/reported previously? If yes, please cite the paper. If not, those results need to be explained here. Currently, the paper begins with testing CF3 for bioactivity.

Response 3: The isolation work was mentioned in 3.1. The strain CF3 was the only bacteria showed inhibitory activity against fungal pathogens among the strains we obtained from our samples. Thus, we did not identify those bacterial strains that did not show any inhibitory activity.

Comment 4: Line 47 - Capitalize 'Root', and

Response4: We have made changes according to the comments.

Comment 5: Line 80 - change to '...planted in banana...'

Response5: We have made changes according to the comments, change to

Comment 6: Line 208 and 299 - change 'in greenhouse condition' to 'under greenhouse conditions'

Response 6: We have changed "in greenhouse condition" to "under greenhouse conditions" to reflect the controlled environment in which the experiments were conducted.

Comment 7: Figure 3 legend, Line 296  - change 'A. lance' to 'A. lancea'

Response 7: We have corrected "A. lancea" to "A. lancea throughout the manuscript to ensure accuracy.

We will revise the manuscript accordingly and submit the updated version for your consideration. Thank you again for your valuable input.

Best regards,

Tielin Wang

Round 2

Reviewer 2 Report

Comments and Suggestions for Authors

Minor editing is required.

Line 160 - change 'stain' to 'strain'

Line 161 - change 'grew' to 'grown'

Comments on the Quality of English Language

Line 114 - sentence is incomprehensible and must be changed.